# Laboratory-based versus population-based surveillance of antimicrobial resistance to inform empirical treatment for suspected urinary tract infection in Indonesia

Adhi Kristianto Sugianli[1], Franciscus Ginting[2], R. Lia Kusumawati[3], Ida Parwati[1], Menno D. de Jong[4], Frank van Leth[5,6], Constance Schultsz[4,5,6] *

1 Department of Clinical Pathology, Faculty of Medicine, Universitas Padjadjaran, Dr. Hasan Sadikin General Hospital, Bandung, West Java, Indonesia, 2 Department of Internal Medicine, Faculty of Medicine, Universitas Sumatera Utara, H. Adam Malik Hospital, Medan, North Sumatera Indonesia, 3 Department of Microbiology, Faculty of Medicine, University of Sumatera Utara, H. Adam Malik Hospital, Medan, Indonesia, 4 Department of Medical Microbiology, Amsterdam University Medical Centers, University of Amsterdam, Amsterdam, the Netherlands, 5 Department of Global Health, Amsterdam University Medical Centers, University of Amsterdam, Amsterdam, the Netherlands, 6 Amsterdam Institute for Global Health and Development, Amsterdam, the Netherlands

☯ These authors contributed equally to this work.
* c.schultsz@aighd.org

**Data Availability Statement:** All relevant data are within the manuscript and its Supporting Information files. A description of the data is

## Abstract

Surveillance of antimicrobial resistance (AMR) enables monitoring of trends in AMR prevalence. WHO recommends laboratory-based surveillance to obtain actionable AMR data at local or national level. However, laboratory-based surveillance may lead to overestimation of the prevalence of AMR due to bias. The objective of this study is to assess the difference in resistance prevalence between laboratory-based and population-based surveillance (PBS) among uropathogens in Indonesia. We included all urine samples submitted to the laboratory growing *Escherichia coli* and *Klebsiella pneumoniae* in the laboratory-based surveillance. Population-based surveillance data were collected in a cross-sectional survey of AMR in *E. coli* and *K. pneumoniae* isolated from urine samples among consecutive patients with symptoms of UTI, attending outpatient clinics and hospital wards. Data were collected between 1 April 2014 until 31 May 2015. The difference in percentage resistance (95% confidence intervals) between laboratory- and population-based surveillance was calculated for relevant antibiotics. A difference larger than +/- 5 percent points was defined as a biased result, precluding laboratory-based surveillance for guiding empirical treatment. We observed high prevalence of AMR ranging between 63.1% (piperacillin-tazobactam) and 85% (ceftriaxone) in laboratory-based surveillance and 41.3% (piperacillin-tazobactam) and 74.2% (ceftriaxone) in population-based surveillance, except for amikacin and meropenem (5.7%/9.8%; 10.8%/5.9%; [laboratory-/population-based surveillance], respectively). Laboratory-based surveillance yielded significantly higher AMR prevalence estimates than population-based surveillance. This difference was much larger when comparing surveillance data from outpatients than from inpatients. All point estimates of the difference between the two surveillance systems were larger than 5 percent points, except for amikacin and

available in the project information on Figshare (https://figshare.com/projects/SPIN_bias_study/73233), which has two items attached: the dataset (doi.org/10.6084/m9.figshare.11378745.v1), and the codebook (doi.org/10.6084/m9.figshare.11379138.v1).

**Funding:** This study was funded through a grant of the Royal Netherlands Academy of Arts and Sciences as part of the Scientific Program Indonesia-the Netherlands (SPIN). The funders had no role in study design, data collection and analysis, decision to publish, or preparation of the manuscript.

**Competing interests:** The authors have declared that no competing interests exist.

meropenem. Laboratory-based AMR surveillance of uropathogens, is not adequate to guide empirical treatment for community-based settings in Indonesia.

## Background

Surveillance of antimicrobial resistance (AMR) enables monitoring of trends in AMR prevalence and is an important tool in the fight against the increasing threat of AMR globally. Surveillance of AMR is needed to inform policy-makers, regulators, and clinicians in support of recommendations for (inter)national policy and local antimicrobial stewardship activities in health facilities, to ultimately reduce AMR associated mortality and morbidity. Low- and middle-income countries (LMIC) are affected disproportionally by the emergence of AMR due to weak national and local policies, lack of quality diagnostic and surveillance capacity, and lack of antibiotic stewardship programs [1].

The WHO Global Action Plan on Antimicrobial Resistance recognizes surveillance as one of its five pillars of action [2]. In WHO's Global Antimicrobial resistance Surveillance System (GLASS), laboratory-based surveillance of AMR is recognized as a priority for the development of strategies to contain antibiotic resistance, and for assessment of the impact of interventions [3]. Laboratory-based surveillance with linkage to patient information is considered as the most efficient and feasible surveillance approach because the data are generated by microbiology laboratories that routinely identify and determine the susceptibility of bacteria isolated from clinical specimens submitted to the laboratory. Population-based surveillance, which is based on surveillance of individuals in a defined population who present with signs and symptoms that meet a clinical case definition, provides more precise data about the burden of AMR in this population. However, population-based surveillance is often considered too laborious and may require resources and capacity that are not available where patients present with symptoms [3]. Whilst laboratory-based surveillance can be used to provide information on local AMR prevalence with the aim to guide the empirical treatment choices, results of laboratory-based surveillance may be biased because of the potential barriers to and selection processes for submission of clinical specimens to laboratories for culture and susceptibility testing, particularly in resource-constrained settings such as in LMIC [3,4]. This bias may result in laboratory-based surveillance results being skewed towards higher prevalence of AMR. Previous studies have assessed the potential sources of bias in laboratory-based surveillance, but studies that assess the actual difference in prevalence estimates between laboratory-based AMR surveillance and population-based surveillance in LMIC are lacking [5].

Indonesia is a lower-middle income country with the world's 4th largest population, where almost all bacteriology laboratories in tertiary hospitals and district laboratories carry out antimicrobial susceptibility testing (AST). Several hospitals report cumulative AST reports every six months, but the AST data are not linked to patient information. Moreover, AST reports are not aggregated at national level, due to difficulties in networking of hospitals, district laboratories and research centers [1,6]. We previously performed a population-based survey of AMR in *Escherichia coli* and *Klebsiella pneumoniae* isolated from patients with symptoms of urinary tract infection (UTI) in Indonesia [7]. Comparing these results to routine laboratory results obtained in the same setting and period allowed us to assess the magnitude of bias of laboratory-based surveillance.

## Materials and methods

### Study design

We compared two surveillance approaches performed in an overlapping time frame in a tertiary referral hospital and in outpatients clinics in Medan. The hospital services the city of Medan as well as the provinces North Sumatera, Aceh, West Sumatera and Riau on the island of Sumatra. We collected laboratory-based AMR surveillance data from 1 April 2014 until 31 May 2015 to coincide with data collected through population-based AMR surveillance in the same time period. Laboratory data were collected retrospectively from the computer-based laboratory records, consisting of routine microbiological investigations on all clinical urine specimens received both from inpatients and outpatients, with a positive culture that yielded *Escherichia coli* and/or *Klebsiella pneumoniae* and their AST results. From patients with multiple positive urine cultures, only the first culture result was included in the study [3,8]. During the surveillance period, systematic screening cultures of urine, for example as part of outbreak management or detection of asymptomatic carriage of (multi-drug resistant) microorganisms, was not performed in the hospital.

Population-based AMR surveillance data were collected in a cross-sectional survey of AMR in *E. coli* and *K. pneumoniae* isolated from urine samples from patients suspected of a UTI, carried out from 1 April 2014 until 31 May 2015, as described previously [7]. In brief, consecutive patients attending four public and private outpatient clinics of urology and obstetrics/ gynaecology, or all patients who were admitted to the internal medicine-, surgery-, obstetrics/ gynaecology-, or neurology wards, were actively screened for the presence of symptoms of UTI, according to CDC definitions [9]. Inpatients were screened for these symptoms on a daily basis. Laboratory procedures were carried out following CLSI guidelines [10].

We included only *E. coli* and *K. pneumoniae* in this study, since those pathogens are the most commonly observed uropathogens, as also recommended in WHO's GLASS [3]. Bacteria which were not identified as *E. coli* or *K. pneumoniae*, we classified as "other" and not included in the direct comparison of the two surveillance strategies.

### Laboratory procedures

**Laboratory-based surveillance.**   Routine microbiological investigations on all clinical urine specimens received, were performed following CLSI guidelines and using in-house standard operating procedures of the hospital microbiology laboratory in Medan [8]. All urine specimens were cultured on blood agar and MacConkey Agar. Any growth on those agar plates was identified to detect uropathogens, using the Vitek2 Compact platform (Biomerieux, France). Uropathogens showing growth of $10^5$ colony forming units (CFUs)/ml or greater were submitted to antimicrobial susceptibility testing (AST) (Vitek AST GN-N317, & GN-N100, Biomerieux) using the same instrument. *Escherichia coli* ATCC 25922 was used as quality control strain for identification and AST [11].

**Population-based surveillance.**   From all included patients, a urine specimen was collected for urinary dipstick analysis. All urine specimens with a positive dipstick test result (positive leukocyte esterase and/or nitrite reaction) were cultured and suspected colonies with growth of $10^3$ CFU/ml or greater and identified to be *E. coli* or *K. pneumoniae* using standard biochemical tests (IMVIC), were submitted to AST using disk diffusion method according to CLSI guidelines [11]. Quality controls (QC) were included for media preparation and QC for susceptibility testing were performed on a weekly basis according to CLSI guidelines, using reference strains *E. coli* ATCC25922, *E. coli* ATCC 35218 and *K. pneumoniae* ATCC 700603 [11].

**Antimicrobial susceptibility tests.** The antimicrobial drugs tested routinely in the microbiology laboratory as well as included in the population-based study, were amoxicillin-clavulanic acid, amikacin, ceftazidime, ceftriaxone, levofloxacin, meropenem and piperacillin-tazobactam [3]. AST results were interpreted as susceptible, intermediate or resistant according to breakpoints from CLSI document M100-S22 for both automated and manual AST [11]. An intermediate test result was considered resistant.

## Data analysis

All datasets were collected and available as electronic files, capturing basic information on patient characteristics (inpatient, outpatient), bacteria isolated, antimicrobial agents tested and inhibitory zone diameter or MIC. The data extraction procedure from the Vitek 2 Compact System was done according to the manufacturer instruction. Isolates were determined susceptible or resistant using CLSI 2012 breakpoints for both laboratory- and population-based surveillance approaches since these were the breakpoints used during the study time period [9,11]. Data were analyzed using Stata 12.1 (Stata Corp, TX, USA).

We determined the percentage points difference in prevalence estimation between the two surveillance approaches and calculated the 95% confidence interval (CI) of this difference, for each antibiotic tested. We arbitrarily considered bias to be present if the point estimate of the difference between the two surveillance approaches was larger than +/- 5 percentage points, on the basis of clinical relevance for empirical treatment guidelines[12].

We first assessed the difference in prevalence estimates between laboratory-based surveillance and population-based surveillance for all isolates. Subsequently we stratified this analysis by inpatient and outpatient settings. We performed a sensitivity analysis for these comparisons with a unified definition of a positive culture as $\geq 10^5$ CFUs/ml of a given pathogen present after growth on MacConkey agar, for both surveillance approaches.

## Ethical approval

This study was approved by the University of Sumatera Utara Faculty of Medicine Ethics Committee, H. Adam Malik General Hospital Research Committee, Universitas Padjadjaran Faculty of Medicine Ethics Committee (286/KOMET/FK USU/ 2013).

## Results

A total of 896 isolates were collected during laboratory-based surveillance of which 474 isolates were *E. coli* or *K. pneumoniae*. Meanwhile, a total number of 645 isolates was collected during the population-based surveillance, of which 508 *E. coli* or *K. pneumoniae* (Table 1).

High prevalence of AMR was observed ranging between 61.7% (piperacillin-tazobactam) and 86.1% (ceftriaxone) in laboratory-based surveillance, and 41.3% (piperacillin-tazobactam) and 74.2% (ceftriaxone) in population-based surveillance. Only for amikacin (6.4% and 9.8% for laboratory-based surveillance and population-based surveillance, respectively), and meropenem (10.9% and 5.9% for laboratory-based surveillance and population-based surveillance, respectively) prevalence estimates were below or around 10% (S1 Table).

Laboratory-based surveillance yielded substantially higher AMR prevalence estimates than population-based surveillance (Fig 1A). This difference was larger when comparing laboratory-based surveillance with population-based surveillance isolates from outpatients than from inpatients (Fig 1B and 1C, S2 and S3 Tables). All point estimates of the difference between the two surveillance approaches were larger than 5 percentage points in the overall analysis and in the outpatient comparison, except for amikacin and meropenem.

**Table 1. Frequency of *Escherichia coli* and *Klebsiella pneumoniae* culture as observed during laboratory- and population-based surveillance from outpatients and inpatients.**

| | Outpatients | | | | Inpatients | | | |
|---|---|---|---|---|---|---|---|---|
| | Laboratory-based N = 227 | | Population-based N = 339 | | Laboratory-based N = 669 | | Population-based N = 306 | |
| | n | % | n | % | n | % | n | % |
| *E. coli* | 124 | 54.6 | 221 | 65.2 | 189 | 28.3 | 199 | 65.0 |
| *K. pneumoniae* | 33 | 14.5 | 40 | 11.8 | 128 | 19.1 | 48 | 15.7 |
| *Other* | 70 | 30.8 | 78 | 23.0 | 352 | 52.6 | 59 | 19.3 |

N = total number of isolates identified; n = total number of isolates per species.

A sensitivity analysis with a unified definition of culture positivity, showed smaller differences in prevalence estimates in the combined inpatient and outpatient analysis and for inpatients only, but showed still marked differences in prevalence estimates for the outpatient population (S4–S6 Tables, S1 Fig).

## Discussion

Both laboratory-based and population-based surveillance approaches showed strikingly high prevalence estimates of AMR for most antibiotics tested, except for amikacin and meropenem. The difference between laboratory-based surveillance estimates and population-based estimates was much larger for outpatients than for inpatients. These differences indicate that laboratory-based AMR prevalence data are not suitable to guide empirical treatment decisions, especially in the outpatient setting.

As recommended by WHO, Indonesia has adopted a National Action Plan to combat AMR in 2017, which includes enhanced surveillance and networking in order to obtain national representative AMR surveillance data to inform guidelines [1]. WHO recommends laboratory-based surveillance with linkage to patient data, as an initial step towards national surveillance since this is considered the most feasible surveillance approach [3]. However, here we show that laboratory-based surveillance is likely to suffer from serious bias, as has been suggested previously [4,13]. Whilst laboratory-based surveillance depends on a clinician's decision to submit a sample for culture and susceptibility testing based on clinical experience or guidelines, potentially leading to differences in case ascertainment and sampling bias, population-based surveillance typically includes all patients who fulfill predefined case definition and inclusion criteria. In LMIC settings where access to diagnostics is often limited due to a range of potential constraints, such selection process may be even more pronounced. Despite Indonesia's progress towards universal health coverage [14], financial constraints may create barriers to bacterial culture and susceptibility testing limiting microbiological diagnostics to those patients with severe or recurrent infections, who often have been pretreated with antibiotics, or to those with insurance coverage that includes diagnostic microbiology, which is often limited to in-patients. This type of selection processes may explain the differences observed between laboratory-based and population-based surveillance in outpatients in the current study. Aggregating laboratory-based AMR surveillance data of UTI outpatients and inpatients has previously been shown to lead to potential overestimation of the prevalence of AMR in outpatients in a high-income setting [4].

We defined the difference between the prevalence estimates as indicating relevant bias at five percent point or more, based on clinical relevance. Whilst this definition is arbitrary, for the outpatient population the difference between laboratory-based and population-based resistance prevalence estimates was much larger than 5 percent points for most antibiotics

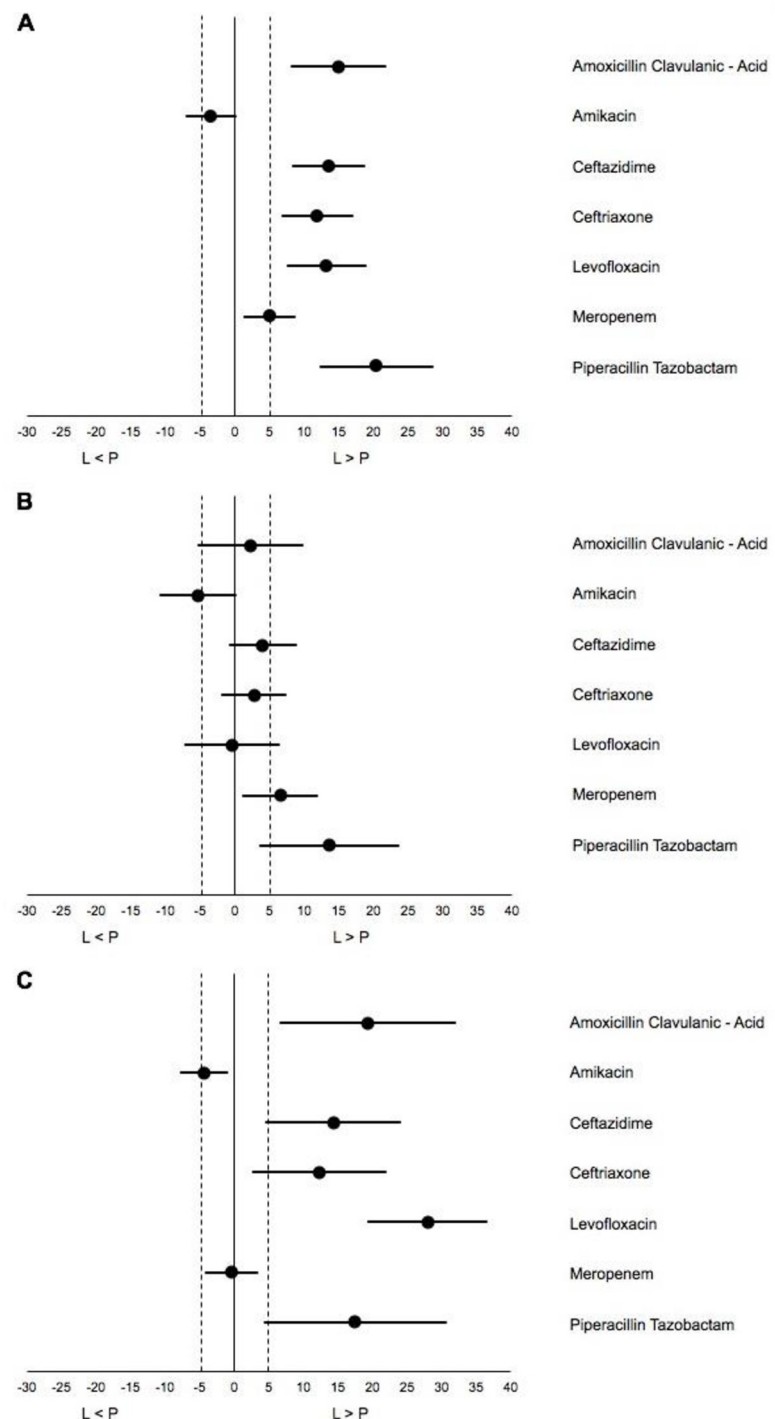

**Fig 1. Difference in prevalence estimates between laboratory- and population-based surveillance.** (A) Total (inpatient & outpatient setting) (B) Inpatients; (C) Outpatients; L>P = Laboratory-based surveillance prevalence estimate of resistance higher than population-based surveillance estimate. Bullets: percentage point difference between laboratory- and population-based surveillance. Horizontal lines: confidence interval for the difference.

analyzed, even when taking into account the uncertainty indicated by the relatively large 95% CIs. We performed a sensitivity analysis which considered differences in the definition of a

positive culture result as a potential source of the observed differences between laboratory-based and population-based surveillance, as also described in previous studies [4,13]. Given that only culture results that are considered clinically relevant lead to a susceptibility test result, such difference between definition of a positive culture may contribute to the differences in prevalence estimates. Indeed, after adjusting the definition of culture positivity in the population-based surveillance to the definition applied in laboratory-based surveillance, a reduction in the difference in prevalence estimates between the two surveillance approaches was observed for almost all antibiotics, however with similar overall conclusions compared to the primary analysis. In this study, we focused our analysis on *E. coli* and *K. pneumoniae* since these are the most common pathogens in UTI and the priority pathogens for surveillance as recommended by WHO-GLASS [3]. However, surveillance of other pathogens may be useful in this setting, in particular for inpatients.

Bias related to differences in gender and age distribution between laboratory-based and population-based surveillance cannot be excluded. Data on gender distribution were not available for the laboratory-based surveillance. Out of 860 samples included in the laboratory-based surveillance, 36 (4%) were submitted from paediatric departments. Only adult patients (age ≥ 18 years) were included in the population-based surveillance. Taken together, a difference in age distribution (paediatric vs adult) is unlikely to explain the differences between laboratory-based and population-based surveillance in this study.

The results of our study indicate that laboratory-based surveillance of uropathogens, in particular when aggregating data of outpatients and inpatients, is likely to overestimate AMR prevalence for outpatient settings. Such overestimation could lead to unwarranted early switch to second line empirical antibiotic treatment in outpatients, which is often more costly and can lead to early emergence of resistance against these second line treatments. Population-based surveillance is more labor intensive and time consuming than laboratory-based surveillance. Alternative strategies should be studied and employed to overcome these drawbacks of conventional population-based surveillance. We have recently shown that using a Lot Quality Assurance Sampling (LQAS) approach is one such alternative strategy for population-based AMR surveillance. Instead of assessment of a prevalence estimate with corresponding confidence intervals, a LQAS-based surveillance approach classifies the prevalence to be above a pre-defined threshold determined on the basis of clinical criteria and guidelines [15]. The study showed that LQAS-based surveillance of the prevalence of AMR provided the opportunity to obtain locally relevant estimation in a timely and affordable manner that can be repeated for monitoring purposes [16]. Other surveillance approaches that provide unbiased population-based AMR prevalence estimates may provide similar solutions and need to be explored.

Our study had some limitations. In the absence of surveillance of nosocomial transmission, we were not informed about potential clonal transmission of urinary pathogens on the hospital wards. However, the routine hospital surveillance report did not show increasing trend of resistance during the study periods, indicating that clonal transmission or hospital outbreaks are unlikely to have affected the results. A second limitation is that the laboratory-based surveillance data were obtained from a single hospital, in contrast to the population based surveillance. However, all data were analysed in the same accredited reference laboratory. Thirdly, different AST methods were used during the two surveillance approaches. Since both methods were performed according to CLSI guidelines using the same breakpoints, this difference is unlikely to affect the surveillance outcomes. Finally, QC performance was done under different protocols. The population-based surveillance was carried out as a research surveillance project with more stringent application of QC procedures, whilst during laboratory-based surveillance standard QC procedures were in place. However, these differences in QC protocols are

unlikely to explain the observed differences in resistance prevalence across all antibiotics studied given the overall direction of high prevalence of resistance, which was similar across the two surveillance approaches.

In conclusion, laboratory-based AMR surveillance of uropathogens, which typically includes a majority of samples from hospital-associated patients, is not adequate to guide empirical treatment for outpatient settings, in Indonesia. Alternative surveillance strategies are needed that provide timely and affordable population-based AMR prevalence data, to inform local and population directed empirical treatment guidelines.

## Supporting information

**S1 Fig. Difference in prevalence estimates between laboratory- and population-based surveillance, unified for definition of culture positivity ($\geq 10^5$ CFU/ml).** (A) Total (inpatient & outpatient setting) (B) Inpatients; (C) Outpatients; L>P = Laboratory-based surveillance prevalence estimate of resistance higher than population-based surveillance estimate. Bullets: percentage difference between laboratory- and population-based surveillance. Horizontal lines: confidence interval for the difference between the two prevalence estimates. Vertical dotted lines indicate to definition of bias (+/-5 percent point difference in prevalence estimate).
(DOCX)

**S1 Table. Difference in resistance prevalence among uropathogens between laboratory-based surveillance and population-based surveillance; combined inpatient and outpatient settings.** Abbrev: n, number of isolates; R, number of resistance isolates; %R, resistance percentage; L, Laboratory-based data; P, Population-based data; %D, Percentage point difference; B, Bias; Y, Yes; N, No; CI, Confidence Interval; lb, lower boundaries; ub, upper boundaries; AMC, Amoxicillin Clavulanic–Acid; AK, Amikacin; CAZ, Ceftazidime; CRO, Ceftriaxone; LVX, Levofloxacin; MEM, Meropenem; TZP, Piperacillin Tazobactam.
(DOCX)

**S2 Table. Difference in resistance prevalence among uropathogens between laboratory-based surveillance and population-based surveillance; inpatient setting.** Abbrev: n, number of isolates; R, number of resistance isolates; %R, resistance percentage; L, Laboratory-based data; P, Population-based data; %D, Percentage point difference; B, Bias; Y, Yes; N, No; CI, Confidence Interval; lb, lower boundaries; ub, upper boundaries; AMC, Amoxicillin Clavulanic–Acid; AK, Amikacin; CAZ, Ceftazidime; CRO, Ceftriaxone; LVX, Levofloxacin; MEM, Meropenem; TZP, Piperacillin Tazobactam.
(DOCX)

**S3 Table. Difference in resistance prevalence among uropathogens between laboratory-based surveillance and population-based surveillance; outpatient setting.** Abbrev: n, number of isolates; R, number of resistance isolates; %R, resistance percentage; L, Laboratory-based data; P, Population-based data; %D, Percentage point difference; B, Bias; Y, Yes; N, No; CI, Confidence Interval; lb, lower boundaries; ub, upper boundaries; AMC, Amoxicillin Clavulanic–Acid; AK, Amikacin; CAZ, Ceftazidime; CRO, Ceftriaxone; LVX, Levofloxacin; MEM, Meropenem; TZP, Piperacillin Tazobactam.
(DOCX)

**S4 Table. Difference in resistance prevalence among uropathogens between laboratory-based surveillance and population-based surveillance; unified for definition of culture positivity ($\geq 10^5$ CFU/ml); combined inpatient and outpatient settings.** Abbrev: n, number of isolates; R, number of resistance isolates; %R, resistance percentage; L, Laboratory-based data;

P, Population-based data; %D, Percentage point difference; B, Bias; Y, Yes; N, No; CI, Confidence Interval; lb, lower boundaries; ub, upper boundaries; AMC, Amoxicillin Clavulanic–Acid; AK, Amikacin; CAZ, Ceftazidime; CRO, Ceftriaxone; LVX, Levofloxacin; MEM, Meropenem; TZP, Piperacillin Tazobactam.
(DOCX)

**S5 Table. Difference in resistance prevalence among uropathogens between laboratory-based surveillance and population-based surveillance; unified for definition of culture positivity ($\geq 10^5$ CFU/ml); inpatient setting.** Abbrev: n, number of isolates; R, number of resistance isolates; %R, resistance percentage; L, Laboratory-based data; P, Population-based data; %D, Percentage point difference; B, Bias; Y, Yes; N, No; CI, Confidence Interval; lb, lower boundaries; ub, upper boundaries; AMC, Amoxicillin Clavulanic–Acid; AK, Amikacin; CAZ, Ceftazidime; CRO, Ceftriaxone; LVX, Levofloxacin; MEM, Meropenem; TZP, Piperacillin Tazobactam.
(DOCX)

**S6 Table. Difference in resistance prevalence among uropathogens between Laboratory-based surveillance and Population-based surveillance; unified for definition of culture positivity ($\geq 10^5$ CFU/ml); outpatient setting.** Abbrev: n, number of isolates; R, number of resistance isolates; %R, resistance percentage; L, Laboratory-based data; P, Population-based data; %D, Percentage point difference; B, Bias; Y, Yes; N, No; CI, Confidence Interval; lb, lower boundaries; ub, upper boundaries; AMC, Amoxicillin Clavulanic–Acid; AK, Amikacin; CAZ, Ceftazidime; CRO, Ceftriaxone; LVX, Levofloxacin; MEM, Meropenem; TZP, Piperacillin Tazobactam.
(DOCX)

## Acknowledgments

We thank Perry Boy Chandra Siahaan and Dewi for laboratory-based data collection; Rohmawati, Elfrina, Sisca, Merlina S. Munthe, Asni Angkat, Mery Heln, Sumarni, Sonti Pangaribuan, Rinawaty Sitepu for their assistance with data collection.

## Author Contributions

**Conceptualization:** Menno D. de Jong, Frank van Leth, Constance Schultsz.

**Formal analysis:** Adhi Kristianto Sugianli, Frank van Leth, Constance Schultsz.

**Investigation:** Adhi Kristianto Sugianli, Franciscus Ginting, R. Lia Kusumawati.

**Methodology:** Frank van Leth, Constance Schultsz.

**Supervision:** R. Lia Kusumawati, Ida Parwati, Menno D. de Jong, Constance Schultsz.

**Writing – original draft:** Adhi Kristianto Sugianli, Frank van Leth, Constance Schultsz.

**Writing – review & editing:** Franciscus Ginting, R. Lia Kusumawati, Ida Parwati, Menno D. de Jong, Frank van Leth.

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
