## [Decision Letter · Decision Letter 0]

12 Nov 2019

PONE-D-19-19644

Laboratory-based versus population-based surveillance of antimicrobial resistance to inform empirical treatment for suspected urinary tract infection in Indonesia

PLOS ONE

Dear Prof. Schultsz,

Thank you for submitting your manuscript to PLOS ONE. After careful consideration, we feel that it has merit but does not fully meet PLOS ONE’s publication criteria as it currently stands. Therefore, we invite you to submit a revised version of the manuscript that addresses the points raised during the review process.

The manuscript has been assessed by two reviewers; their comments are available below.

The reviewers find the work of relevance but have raised some comments that need attention in a revision. The reviewers recommend that the relationship to the study reported in your earlier publication in J Antimicrob Chemother. 2017;72: 1469–77 is described in greater detail and in particular, that you clarify any overlap in sample populations between the two studies. The reviewers recommend that the analysis of additional pathogens is included, or if that is not possible, that this is clearly acknowledged as a limitation.

Could you please revise the manuscript to carefully address the concerns raised by the reviewers?

We would appreciate receiving your revised manuscript by Dec 26 2019 11:59PM. To enhance the reproducibility of your results, we recommend that if applicable you deposit your laboratory protocols in protocols.io, where a protocol can be assigned its own identifier (DOI) such that it can be cited independently in the future. For instructions see: http://journals.plos.org/plosone/s/submission-guidelines#loc-laboratory-protocols

We look forward to receiving your revised manuscript.

Kind regards,

Iratxe Puebla

Senior Managing Editor, PLOS ONE

Journal Requirements:

Reviewers' comments:

Reviewer's Responses to Questions

**Comments to the Author**

1. Is the manuscript technically sound, and do the data support the conclusions?

Reviewer #1: Yes

Reviewer #2: Yes

2. Has the statistical analysis been performed appropriately and rigorously? 

Reviewer #1: Yes

Reviewer #2: Yes

3. Have the authors made all data underlying the findings in their manuscript fully available?

Reviewer #1: Yes

Reviewer #2: Yes

4. Is the manuscript presented in an intelligible fashion and written in standard English?

Reviewer #1: Yes

Reviewer #2: Yes

5. Review Comments to the Author

Reviewer #1: Overview:

The study assessed the difference in antibiotic resistance between laboratory based surveillance and population based surveillance in uropathogens in Indonesia. They focused the microbiology work on Escherichia coli and Klebsiella pneumoniae as these are the main burden of UTIs. The population based surveillance involved collection through a cross sectional survey from people with symptoms corresponding to UTI attending clinics and hospitals as outpatients. Data was collected between the 1st of April 2014 and the 31st of May 2015. Some of the main results were that there was a higher prevalence of resistance to Piperacillin-tazobactam (63.1%) and ceftriaxone (85.2%) in the laboratory based surveillance strains compared to the population based surveillance at 41.3% and 74.2% respectively.

Overall comments:

This is a very relevant study which addresses many questions being raised at the moment to understand the bias of results reported in hospital microbiology laboratories. A community based study such as this will help to solve many questions on bias for antibiotic resistance.

Specific queries:

• Although I note this study was previously published and referenced in the current publication [1], I believe there needs to be more description about the study subjects. In particular I would like to understand how the researchers were able to determine the patients who were in patients in the population based study. Particularly as the major difference between the laboratory and population based studies occurred in the outpatients when the population based inpatients were removed. Further explanation about this is necessary in the text.

• Please further explain the following sentence in your discussion: “A reduction in resistance was observed when only clinically relevant culture results were reported in the laboratory”, what does this mean?

• It would be useful to have a map of the area depicting where the hospital is based and where the population based work took place

• What is the age and gender distribution of your patients? Could there be a difference in your results due to either of these variables?

• The major difference seems to be between outpatient samples and whether the results are derived from the laboratory or the population based work, although numbers of strains are low in these groups for Klebsiella pneumoniae at 33 and 40 strains respectively). I would like to see more discussion on your thoughts as to why this is the case.

• Is there a difference in prescribing of antibiotics in those patients who are included in the population based study compared to the laboratory results, might this impact on the results that you have reported?

Reference

1. Sugianli, A.K., et al., Antimicrobial resistance in uropathogens and appropriateness of empirical treatment: a population-based surveillance study in Indonesia. J Antimicrob Chemother, 2017. 72(5): p. 1469-1477.

Reviewer #2: The authors have assessed laboratory-based surveillance versus population-based surveillance for studying the prevalence of antimicrobial resistance in UTI in Indonesia. The data presented is sound and the methodology as well as the analysis are supporting the outcome of this paper. I have a few minor suggestions that may improve the paper further:

MINOR:

- In the background, the authors address the bias in the selection process that may occur in laboratory based surveillance. I would suggest adding some clarification on the reasoning behind this bias.

- Is the population-based surveillance data presented in this manuscript exactly as the same data that was previously published in J Antimicrob Chemother. 2017;72: 1469–77?

If so, I would suggest including a clarification to further highlight this point. I also would like to encourage the authors to visit the permission requirements of JAC to make sure that the reuse of the published data is in alignment with the journal's policy. This can be found in https://academic.oup.com/jac/pages/General_Instructions#Permissions

MAJOR:

- The authors have clearly addressed the possible bias that may occur from laboratory-based surveillance, alongside the selection bias on E. coli and K. pneumoniae to represent uropathogens, as suggested by GLASS. When conducting this study, the authors have decided to apply to same bias and only select for E. coli and K. pneumoniae in the population based study. Is it possible to include the analysis of the other pathogens isolated in the results? If not possible, I think it would be important to address this limitation and highlight the need to test the value of population-based surveillance on a wider range of uropathogens.

6. PLOS authors have the option to publish the peer review history of their article (what does this mean?). If published, this will include your full peer review and any attached files.

Reviewer #1: Yes: Catrin E Moore

Reviewer #2: Yes: Hosam Zowawi

---

## [Author Response · Author response to Decision Letter 0]

9 Jan 2020

Reviewer #1: Overview:

The study assessed the difference in antibiotic resistance between laboratory based surveillance and population based surveillance in uropathogens in Indonesia. They focused the microbiology work on Escherichia coli and Klebsiella pneumoniae as these are the main burden of UTIs. The population based surveillance involved collection through a cross sectional survey from people with symptoms corresponding to UTI attending clinics and hospitals as outpatients. Data was collected between the 1st of April 2014 and the 31st of May 2015. Some of the main results were that there was a higher prevalence of resistance to Piperacillin-tazobactam (63.1%) and ceftriaxone (85.2%) in the laboratory based surveillance strains compared to the population based surveillance at 41.3% and 74.2% respectively.

Overall comments:

This is a very relevant study which addresses many questions being raised at the moment to understand the bias of results reported in hospital microbiology laboratories. A community based study such as this will help to solve many questions on bias for antibiotic resistance.

We thank the reviewer for the positive comments.

Specific queries:

• Although I note this study was previously published and referenced in the current publication [1], I believe there needs to be more description about the study subjects. In particular I would like to understand how the researchers were able to determine the patients who were in patients in the population based study. Particularly as the major difference between the laboratory and population based studies occurred in the outpatients when the population based inpatients were removed. Further explanation about this is necessary in the text.

We have described the inclusion of inpatients in the Methods section:

“Population-based AMR surveillance data were collected in a cross-sectional survey of AMR in E. coli and K. pneumoniae isolated from urine samples from patients suspected of a UTI, carried out from 1 April 2014 until 31 May 2015, as described previously. In brief, consecutive patients attending four public and private outpatient clinics of urology and obstetrics/gynaecology, or who were admitted to the internal medicine-, surgery-, obstetrics/gynaecology-, or neurology wards, were actively screened for the presence of symptoms of UTI, according to CDC definitions.” 

In other words, the population of inpatients included in the population-based study consisted of those who were admitted to the internal medicine-, surgery-, obstetrics/gynaecology-, or neurology wards at the time of sampling and had symptoms of UTI. Admitted patients were screened on a daily basis for these symptoms. We have added more details regarding the inclusion procedure of inpatients to the Methods section.

• Please further explain the following sentence in your discussion: “A reduction in resistance was observed when only clinically relevant culture results were reported in the laboratory”, what does this mean?

We are not clear which sentence the reviewer is referring to. The sentence the reviewer is quoting is not in our manuscript.

• It would be useful to have a map of the area depicting where the hospital is based and where the population based work took place

All study sites are in the city of Medan. The Adam Malik hospital is a tertiary referral hospital which services the city of Medan as well as the provinces North Sumatera, Aceh, West Sumatera and Riau on the island of Sumatra. We have provided this information which we believe is more helpful than a city map of Medan.

 • What is the age and gender distribution of your patients? Could there be a difference in your results due to either of these variables?

Data on gender and age distribution were not available for the laboratory-based surveillance. However, out of 860 samples, 36 (4%) were from paediatric departments. Only adult patients (≥ 18 years) were included in the population-based surveillance. Taken together, the difference in gross age distribution (paediatric vs adult) is unlikely to explain the differences between laboratory-based and population-based surveillance. We have added this information to the Discussion.

• The major difference seems to be between outpatient samples and whether the results are derived from the laboratory or the population based work, although numbers of strains are low in these groups for Klebsiella pneumoniae at 33 and 40 strains respectively). I would like to see more discussion on your thoughts as to why this is the case.

As indicated in the Discussion, the major difference in AMR estimates between the two surveillance approaches is explained by the fact that in laboratory-based surveillance prevalence estimates for outpatients are markedly higher than estimates in the population-based surveillance. We have elaborated on the potential causes of this difference in the Discussion and consider sampling bias in the laboratory-based surveillance (clinician’s decision to submit a sample to the laboratory vs systematic inclusion), laboratory practice (definition of a positive culture), as well as differences in age distribution (see comment above). We focused our analysis on E. coli and K. pneumoniae since these are the most common pathogens in UTI and since these are the priority pathogens for surveillance as recommended by WHO-GLASS. We analysed these two pathogens together and did not perform a separate analysis for the K. pneumoniae isolates since this is compatible with clinical practice where during prescription of empirical therapy the causative pathogen is unknown but likely to be E. coli and/or K. pneumoniae. We have modified the Discussion to include the latter consideration.

• Is there a difference in prescribing of antibiotics in those patients who are included in the population based study compared to the laboratory results, might this impact on the results that you have reported?

The primary objective of our study was to assess the difference in AMR prevalence estimates between laboratory-based and population-based surveillance. We did not specifically focus on the antibiotic pre-treatment or prescriptions. Indeed, differences in antibiotic pre-treatment may explain some of the bias which we observe in laboratory-based surveillance because patients for whom samples have been submitted may have been pre-treated more frequently. We have added this to the description of the bias in laboratory-based surveillance in the Discussion.

Reviewer #2: The authors have assessed laboratory-based surveillance versus population-based surveillance for studying the prevalence of antimicrobial resistance in UTI in Indonesia. The data presented is sound and the methodology as well as the analysis are supporting the outcome of this paper. I have a few minor suggestions that may improve the paper further:

We thank the reviewer for the positive comments.

MINOR:

- In the background, the authors address the bias in the selection process that may occur in laboratory based surveillance. I would suggest adding some clarification on the reasoning behind this bias.

We added some examples of potential bias in laboratory-based surveillance to clarify.

- Is the population-based surveillance data presented in this manuscript exactly as the same data that was previously published in J Antimicrob Chemother. 2017;72: 1469–77?

If so, I would suggest including a clarification to further highlight this point. I also would like to encourage the authors to visit the permission requirements of JAC to make sure that the reuse of the published data is in alignment with the journal's policy. This can be found in https://academic.oup.com/jac/pages/General_Instructions#Permissions

The data are not exactly the same; we used a limited data set, i.e. only one of two study areas from our previous study that matched with the laboratory-based surveillance data which we added to this study. We checked the JAC requirements.

MAJOR:

- The authors have clearly addressed the possible bias that may occur from laboratory-based surveillance, alongside the selection bias on E. coli and K. pneumoniae to represent uropathogens, as suggested by GLASS. When conducting this study, the authors have decided to apply to same bias and only select for E. coli and K. pneumoniae in the population based study. Is it possible to include the analysis of the other pathogens isolated in the results? If not possible, I think it would be important to address this limitation and highlight the need to test the value of population-based surveillance on a wider range of uropathogens.

We focused our analysis on E. coli and K. pneumoniae since these are the most common pathogens in UTI and since these are the priority pathogens for surveillance as recommended by WHO-GLASS. We therefore do not consider our focus as a limitation of the study. We don’t have additional information for other pathogens isolated for the population-based surveillance. However, we agree that for inpatients in particular, surveillance of other pathogens may be helpful. We added this to the Discussion.

Authors’ note added to the review: 

- We have corrected the denominators in S4-6 Tables and S2 Figure. These corrections do not affect the results.

- We have added the URL at which the study data and associated code book can be accessed.

---

## [Decision Letter · Decision Letter 1]

3 Mar 2020

Laboratory-based versus population-based surveillance of antimicrobial resistance to inform empirical treatment for suspected urinary tract infection in Indonesia

PONE-D-19-19644R1

Dear Dr. Schultsz,

We are pleased to inform you that your manuscript has been judged scientifically suitable for publication and will be formally accepted for publication once it complies with all outstanding technical requirements.

With kind regards,

Davida S. Smyth, Ph.D.

Academic Editor

PLOS ONE

Additional Editor Comments (optional):

Reviewers' comments:

Reviewer's Responses to Questions

**Comments to the Author**

1. If the authors have adequately addressed your comments raised in a previous round of review and you feel that this manuscript is now acceptable for publication, you may indicate that here to bypass the “Comments to the Author” section, enter your conflict of interest statement in the “Confidential to Editor” section, and submit your "Accept" recommendation.

Reviewer #1: All comments have been addressed

2. Is the manuscript technically sound, and do the data support the conclusions?

Reviewer #1: Yes

3. Has the statistical analysis been performed appropriately and rigorously? 

Reviewer #1: Yes

4. Have the authors made all data underlying the findings in their manuscript fully available?

Reviewer #1: Yes

5. Is the manuscript presented in an intelligible fashion and written in standard English?

Reviewer #1: Yes

6. Review Comments to the Author

Reviewer #1: The authors have addressed my comments adequately, I have no additional comments on the new manuscript.

7. PLOS authors have the option to publish the peer review history of their article (what does this mean?). If published, this will include your full peer review and any attached files.

Reviewer #1: Yes: Dr Catrin Moore

---

## [Editor Report · Acceptance letter]

13 Mar 2020

PONE-D-19-19644R1 

Laboratory-based versus population-based surveillance of antimicrobial resistance to inform empirical treatment for suspected urinary tract infection in Indonesia 

Dear Dr. Schultsz:

I am pleased to inform you that your manuscript has been deemed suitable for publication in PLOS ONE. Congratulations! Your manuscript is now with our production department. 

With kind regards,

on behalf of

Dr. Davida S. Smyth 

Academic Editor

PLOS ONE